# Cold-Adapted Proteases: An Efficient and Energy-Saving Biocatalyst

**DOI:** 10.3390/ijms24108532

**Published:** 2023-05-10

**Authors:** Zhengfeng Yang, Zhendi Huang, Qian Wu, Xianghua Tang, Zunxi Huang

**Affiliations:** 1Key Laboratory of Yunnan for Biomass Energy and Biotechnology of Environment, Yunnan Normal University, Kunming 650000, China; yzfwym110803@163.com; 2School of Life Sciences, Yunnan Normal University, Kunming 650000, China; zhendi@163.com (Z.H.); qian99223@163.com (Q.W.); txhdom@163.com (X.T.); 3Key Laboratory of Enzyme Engineering, Yunnan Normal University, Kunming 650000, China; 4Engineering Research Center of Sustainable Development and Utilization of Biomass Energy, Ministry of Education, Yunnan Normal University, Kunming 650000, China

**Keywords:** cold-adapted proteases, green energy conservation, stability, medical applications

## Abstract

The modern biotechnology industry has a demand for macromolecules that can function in extreme environments. One example is cold-adapted proteases, possessing advantages such as maintaining high catalytic efficiency at low temperature and low energy input during production and inactivation. Meanwhile, cold-adapted proteases are characterised by sustainability, environmental protection, and energy conservation; therefore, they hold significant economic and ecological value regarding resource utilisation and the global biogeochemical cycle. Recently, the development and application of cold-adapted proteases have gained gaining increasing attention; however, their applications potential has not yet been fully developed, which has seriously restricted the promotion and application of cold-adapted proteases in the industry. This article introduces the source, related enzymology characteristics, cold resistance mechanism, and the structure-function relationship of cold-adapted proteases in detail. This is in addition to discussing related biotechnologies to improve stability, emphasise application potential in clinical medical research, and the constraints of the further developing of cold-adapted proteases. This article provides a reference for future research and the development of cold-adapted proteases.

## 1. Introduction

Proteases are a class of enzymes that can hydrolyse large proteins or peptides into amino acids or small peptides. The small peptides generated by hydrolysis often have unique physiological activity and are used in medicine, health care, and skin care products. Initial research on proteases can be traced back to the early 20th century; since then, with the rapid development of biotechnology, many researchers conducted comprehensive studies on proteases, making significant progress regarding strain screening, mechanism of action, structure-activity relationship, and the heterologous expression of protease. However, most previous research has focused on mesophilic proteases, while few studies have investigated cold-adapted proteases, which were mainly developed in the last 30 years. Cold-adapted proteases are commonly found in animals, plants, and microorganisms that are adapted to low-temperature environments. Most cold-adapted proteases have the optimal catalytic temperature of 10–40 °C and an optimal pH of 8–10. These alkalophilic proteases that can maintain high catalytic efficiency under low and medium temperatures and weakly alkaline conditions [1]. Approximately 80% of the earth’s biosphere and 90% of the marine environment experience temperatures below 5 °C [2]. The cold-adapted proteases used in industry mainly originate from extreme environments (Figure 1), including cold-adapted microorganisms in deep sea waters, polar regions, high mountains and plateaus, and cold deserts [1]. After long-term exposure to low temperature, cold adaptation microorganisms have evolved corresponding cold-adapted mechanisms, such as increased structural flexibility of enzymes, unique cell membrane lipid components, rapid cryoprotectant synthesis mechanisms, and high cold-shock protein expression through environmental adaptation and evolution, to ensure the fluidity of cell membranes and cell folding at low temperatures [3,4]. Compared with mesophilic proteases, cold-adapted proteases have a higher specific activity at low temperatures and require very low activation energies.

Recently, researchers have discovered that psychrophilic microorganisms and cold-adapted enzymes have significant biotechnological potential. They provide many economic and ecological advantages compared with organisms and their enzymes that work at higher temperatures. For example, cold-adapted microorganisms and their cold-adapted proteins and enzymes have many biotechnological applications, in industries such as medicine, washing, textiles, food, waste management, and skin care (Figure 1) [5]. Cold-adapted enzymes can be produced without heating, savings in energy and money. In industrial applications, the high activity of cold-adapted enzymes helps shorten the reaction times without a reduction in efficiency or auxiliary heating, thus also reducing the energy consumption rate [6]. For example, during leather production, cold-adapted proteases do not require additional heating and temperature adjustment, functioning at tap water temperatures compared with traditional mesophilic proteases. As a washing ingredient, cold-adapted proteases as a washing ingredient, this can be used to achieve decontamination at low temperatures, reducing the cost of environmental heating and avoiding possible colour damage resulting from high-temperature washing conditions [7]. In cold environments, the degradation ability of endogenous microorganisms decreases with decreasing temperature, which is not conducive to waste management in cold regions. However, cold-adapted proteases can compensate for this disadvantage. For example, treating organic wastes from fisheries and aquaculture can not only avoid serious health and environmental problems but can also recycle high-value biological molecules within waste sustainably [6].

Notably, proteases, as recognised therapeutic drugs with good tolerance, have been used in medicine for over a decade. Additionally, more than 100 years ago, proteases were used to treat chronic surface ulcers, tubulous lymphadenitis, and tubulous fibrosis [8,9,10]. Proteases are primarily used for treatment in four fields: oral medicine for gastrointestinal diseases, anti-infective drugs, thrombolytic drugs for thromboembolic diseases, and local drugs for wound debridement [11]. The majority, including thrombolytic drugs and coagulants, are used to treat blood diseases. Other approved protease therapies can be applied for digestion, muscle spasms, cosmetics, etc. such as using local proteases for selective tissue destruction to treat skin diseases (such as skin warts or actinic keratosis) [8].

Significant progress has recently been made in the research and application of industrial biocatalysis technology, although the potential of biocatalysis using extreme enzymes has not yet been fully realised [12]. Cold-adapted proteases are also more sensitive to thermal inactivation, low pH, and autolysis [13] while showing lower stability than mesophilic proteases in medium- and high-temperature environments. Improving the application and efficiency of cold-adapted proteases and their stability in medium- and high-temperature environments will be a primary direction for future research. Environmental problems such as climate change and global warming caused by greenhouse gas emissions call for humans to establish a low-carbon and sustainable green economy. Cold-adapted proteases are receiving increasing attention, and their related research, exploration, and application potential are also becoming more important. Therefore, in this paper, cold-adapted proteases sources and mining, classification, enzymology characteristics, low-temperature resistance mechanisms, stability improvement, and industrial applications, including medical applications, are reviewed in detail, while its application potential as a new green and energy-saving catalyst is emphasised.

## 2. Sources and Mining of Cold-Adapted Protease

It was previously believed that extremely cold areas were sterile and thus did not contain any life. Additionally, there was little interest in psychrophilic microorganisms that could grow at low temperatures. However, many microorganisms provide valuable molecular sources with high biotechnological potential [14]. As early as 1975, Morita defined psychrophilic bacteria based on their optimal growth temperatures. Their optimal growth temperature was determined to be 15 °C or lower, while the highest growth temperature was below 20 °C, and they could even grow when exposed to temperatures below 0 °C, differing from ordinary bacteria, whose optimum growth temperature is generally 20–4 °C [15,16]. During long-term exposure to low temperatures, cold-adapted microorganisms have evolved corresponding cold adaptation mechanisms, including increased structural flexibility of enzymes such as proteases, unique cell membrane lipid components, rapid cryoprotectant synthesis, and high cold-shock protein expression [7]. Cold temperatures are known to limit the movement of plants and animals. Thus, microorganisms play a dominant role in the cold environment, and their biological activities also maintain nutrient flow in cold environments while contributing to the global biogeochemical cycle [17].

Cold-adapted proteases primarily originate from cold-adapted microorganisms, which are generally found in extreme environments, such as water and mud in the deep sea, ice and frozen soil in polar regions, glaciers, mountains, and plateaus. The polar region covers an area of approximately 15%, with temperatures below 0 °C throughout the year. Glacial permafrost accounts for approximately 20% of the land area, with an average temperature below 4 °C. The oceans cover more than 70% of the earth, with an average temperature of 5 °C. These extreme environments are important habitats for cold-adapted microorganisms, which can be primarily divided into the following types: Gram-negative bacteria (*Pseudoalteromonas*, Moraxella, psychrophilic bacteria, *Polar monospora*, cold-resistant bacteria, etc.), gram-positive bacteria (*Arthrobacter*, *Bacillus*, *Micrococcus*, and *Archaea*, such as *Methanogens*, *Rhodopseudomonas salina*), yeast (*Candida*, *Cryptococcus*), and fungi (*Penicillium*, *Cladosporium*) [7].

The Himalayas are one of the most representative areas of low-temperature environments on Earth, with its bacterial diversity being recognised for its potential to produce low-temperature active enzymes. It has also been reported that there are temperature-sensitive and -tolerant bacterial isolates in the Qomolangma Glacier Site, indicating rich diversity even in the most extreme and harsh areas [17]. Furthermore, fungi on high-altitude glaciers are strong candidates for biotechnological applications, and according to a survey of the diversity of glacier fungi in Tirich Mir of the Dukush mountains in northern Pakistan, *Penicillium* is the most common, followed by *Alternaria*, which shows highly efficient low-temperature enzyme activity [18]. Park et al. [19] found 15,696 strains of bacteria that had been isolated from 633 seawater samples collected from the Chukchi Sea. Of these, 2526 strains (approximately 16%) were found to have protease activity after screening at a low temperature (15 °C). After 16S rRNA identification, these were found to have been primarily related to the following taxa: *Alteromonas* (31%), *Staphylococcus* (27%), *Pseudoalteromonas* (14%), *Leeuwenhoecella* (7%), *Bacillus* (5%), *Thiobacillus* (5%), cold-resistant bacilli (4%), *Clostridium* (2%), *Acinetobacter* (2%), *Pseudomonas* (1%), *Halomonas* (1%), and *Dukes* (1%). Additionally, Kim et al. [20] isolated 89 strains of bacteria from marine and land samples from the Svalbard Islands, Norway. Of these, 48 strains (approximately 54%), including *Pseudoalteromonas* (33 strains), *Pseudomonas* (10 strains), *Arthrobacter* (4 strains), and *Flavobacterium* (1 strain), were screened at low temperatures and observed to have protease activity.

Recently, many psychrophilic enzymes, including psychrophilic proteases, have been discovered through different biomolecular and genetic engineering technologies. These have good application potential for different industrial applications and include microbial metagenomes from cryogenic environments, psychrophilic enzymes obtained from various cold environments, including proteases [20] and cellulases [21,22], xylanases [23], amylases [24], lipases [25], chitinases [26], pectinases [27], esterases [28,29], and laccases [30]. Three proteases and other enzymes of important commercial value from Antarctic soil were previously obtained using the metagenomic method [31]. Fan et al. successfully obtained a new cold-adapted N-acyl homoserine lactoesterase from the macro microbe genome of musty tofu, which could significantly reduce the production of virulence factors and biofilms of *Pseudomonas aeruginosa* PAO1 and also has the potential to become a candidate drug for treating *P. aeruginosa* infections [32]. Furthermore, Tchigvintsev et al. successfully screened several carboxylesterases from the marine metagenome, including some with excellent cold adaptation and salt tolerance [33].

Metagenomics, as a mining method for discovering new enzyme genes in microorganisms and their communities, bypasses the technical challenge of cultivating polar microorganisms and has shown excellent application potential in many related reports. Screening environmental microbial strains is also an effective method for mining enzyme genes. This method can obtain enzyme genes with special functions through the enrichment and screening of culturable microorganisms in the corresponding environment and is characterised by being fast, simple, and suitable for large-scale screening. Arnórsdottir et al. previously reported that a cold-adapted protease Vpr had been obtained from a marine psychrophilic vibrio, and high expression was achieved using the T7 system of *Escherichia coli* [34,35]. Additionally, Lario et al. studied a protease from cold-adapted yeast from Antarctica before purifying and sequencing it [36]. A protease produced by cold nutritious *Bacillus fetida* was isolated from Himalayan glacial soils. Through qualitative and quantitative screening, the cold-active protease Apr-BO1 was purified, which can effectively remove stains during low-temperature washing and is considered a good detergent additive [37]. Daskaya-Dikmen et al. screened yeast strains of different genera from a cold environment in northern Turkey and screened for cold-adapted pectinases, proteases, amylases, etc. Among them, the production of cold-active amylase by *Cystosporium capitatum* and pectinase by *Rhodosporomyces* was first reported [38]. The comparative study of functional genomics is another efficient and fast method of gene mining. With the progress of high-throughput sequencing technology, the number of publicly available bacterial genomes has increased significantly. Through the comparative analysis of genome sequences, we found genes encoding biological molecules and products of interest. Perfumo et al. used the comparative genomics of Psychrobacter genes to specifically identify and clone a gene that encoded a cold-active protease in isolate 94–6 PB and subsequently characterised the physical and chemical properties of the expression enzyme through in vitro analysis [39].

## 3. Classification of Cold-Adapted Proteases

The conventional classification method for proteases can also be applied to classify cold-adapted proteases. Proteases can be divided into acidic, alkaline, and neutral categories according to the degree of acid–base preference. According to the method of peptide bond hydrolysis, these can be divided into endopeptidases (hydrolysis of internal peptide bonds) and exopeptidases (hydrolysis from the end). Endopeptidases are widely used in industries, depending on the chemical properties of functional groups in catalytic or active sites and can be generally divided into serine proteases, metalloproteinases, cysteine proteases, and aspartate proteases. They can be further classified according to the degree of inhibition of enzyme inhibitors [21] into serine protease inhibitors (benzene sulfonyl fluoride PMSF), metalloproteinase inhibitors (1,10-phenanthroline and EDTA), cysteine protease inhibitors (N-ethylmaleimide), and aspartic protease inhibitors (pepsin inhibitor A). PMSF can covalently bind with serine residues to cause conformational changes in the enzyme molecule; the more serine residues in the active part of the protease, the stronger the inhibition of PMSF on the enzyme. Serine proteases are widely distributed across various taxa, indicating that serine proteases are crucial for the survival of microorganisms [40].

If the protease activity decreases by more than 20% after treatment with inhibitors, which constitutes a significant negative impact, the protease can be classified according to the degree of inhibition. Inhibitor analysis of proteases at low temperatures can determine the type of cold-adapted protease. PMSF reduced the activity of ArcP02, ArcP08, and ArcP11 protease crude enzyme solutions by 24.8%, 39.8%, and 31.4%, respectively, whereas those of ArcP02 and ArcP08 decreased by 27.3% and 21.2%, respectively, at the same concentration of 1,10-phenanthroline. Therefore, it can be considered that the proteases produced by ArcP02, ArcP08, and ArcP11 are serine proteases, whilst ArcP02 and ArcP08 may produce additional metalloproteinases [19]. Table 1 shows that a large proportion of the reported cold-adapted proteases belong to serine proteases [12,40,41,42,43,44,45,46,47,48,49,50], and a part of them belongs to metalloproteinases [51,52]. According to the optimal pH, most cold-adapted proteases belong to neutral proteases [40,50,51,53,54,55] and alkaline proteases [41,42,43,44,45,46,56,57,58,59]. In contrast, very few belong to acid proteases [12,60] because the activity of acid cold-adapted proteases contains aspartic acid residues; therefore, they are also aspartic proteases. In general, cold-adapted proteases are known to be active against various natural proteins and different types of natural and synthetic substrates, showing a wide range of substrate specificity. The cold-adapted serine proteases produced by *Chryseobacterium* sp. have the highest hydrolytic activity against casein, followed by gelatin and bovine serum albumin, and the lowest on egg albumin [40]. The cold-adapted serine peptidase produced by *Lysobacter* sp. has the highest activity against azocasein, followed by gelatin and feather powder, and the lowest activity against casein, bovine serum albumin, and azo keratin [52]. The broad substrate specificity of cold-adapted proteases is of great value in industrial applications, especially in bioremediation processes performed at low temperatures.

## 4. Enzymatic Characteristics of Cold-Adapted Proteases

For normal growth adaptation to life under low-temperature conditions, cold-adapted microorganisms have evolved a set of gene expression sequences and special molecular mechanisms specific to low-temperature environments, in which the high expression of low-temperature active enzymes plays a key role during the process of cold adaptation. In recent decades, research on cold-adapted proteases has initially focused on exploring enzymatic characteristics, including the optimal temperature, pH, and Effects of metal ions (e.g., Hg^2+^, Cd^2+^, Co^2+^, Mn^2+^, Ca^2+^, K^+^, Na^+^, Fe^2+^, etc.) and compounds (such as SDS, PMSF, EDTA, LAS, etc.) on enzyme activity (Table 1). Their optimal catalytic temperature is 10–60 °C, whilst their optimal pH is usually in the neutral to alkaline pH range of 7–10 [1]. To date, thermostable alkaline proteases primarily originate from *Bacillus subtilis* and *Bacillus licheniformis*, whilst proteases from *Geomyces panorum*, and yeast *Candida humicola* are rare acid thermostable proteases with an optimal pH of 3.0 [1,56,60]. Presently, the reported molecular weight of cold-adapted proteases is small, typically in the range of 25–60 kDa; only a few cold-adapted proteases have large molecular weights [39,50,51,57]. Organic reagents such as EDTA, PMSF, and DFP often inhibit the activity of cold-adapted protease [41,42,43], as do metal ions Hg^2+^, Cu^2+^ and Co^2+^ [40,66]. Metal ions, such as Mg^2+^, Mn^2+^, and Ca^2+^ can improve the activity of cold-adapted proteases to varying degrees [46,58].

## 5. Structural Adaptations of Cold-Adapted Protease

The structure of cold-adapted proteases can be analysed from their molecular (amino acid composition and sequence, terminal composition, disulphide bond position, etc.) and three-dimensional structure. Nuclear magnetic resonance spectroscopy, site-directed mutagenesis, X-ray crystal diffraction, and freeze electron microscopy are effective methods that can be used to understand the crystal structure. Additionally, the high structural homology of enzymes at different temperatures provides an opportunity to study the structural characteristics of their adaptation to different temperatures. Through comparative analysis of the structural characteristics of cold-adapted, mesophilic, and thermophilic proteases (Figure 2), as well as previous studies, we can understand how proteases become adapted to low temperatures. These groups comprise protein surfaces that determine the important interactions between proteins and water, which is expected to significantly impact protein function in adapting to high and low temperatures. In many thermophilic proteins, a larger proportion of non-polar surfaces is believed to help increase stability [67,68]. The proportion of serine residues in cold-adapted enzyme molecules of cold-adapted microorganisms is high, while the proportion of proline and acid residues is low. Owing to the high number of polar amino acid residues, these types of enzymes have a more hydrophilic interaction with solvents [49,69]. Therefore, the active sites of these enzymes are usually more accessible to compensate for substrate diffusion at lower temperatures. The difference between cold-adapted Vibrio proteases and more stable proteases is their strong anionic properties due to the numerous uncompensated negatively-charged residues on their surface. In some studies, the difference in surface charge distribution or the increase in the non-polar surface area has been considered to enable adaptation to low temperatures [70,71,72]. On the other hand, the weakness of non-covalent interactions (such as salt bridges, hydrophobic interactions, aromatic interactions, main chains, side chains, and side chain hydrogen bonds) in some special positions can increase the core volume of enzyme molecules and make the whole molecule looser [46], improving the conformational flexibility of proteases [37]. However, the relationship between the number of non-covalent bonds and the temperature adaptation of enzymes cannot be determined at present. For example, the number of salt bridges in cold-adapted protease 1SH7, thermophilic protease 1IC6, and thermophilic protease 1THM was compared [73]. The number of salt bridges in 1SH7 and 1IC6 was the same and was only two less than that in thermophilic protease 1THM. Although there have been many reported instances of salt bridges and hydrogen bonds introduced in the modification of thermal stability enhancement of many enzymes, an important aspect of their contribution to protein stability lies in their location and distribution [73]. However, this increase in flexibility is often accompanied by decreased stability. Cold-adapted enzymes generally show low thermal stability, short half-life, and high sensitivity to low pH [74]. In contrast, thermostable enzymes have a more rigid and compact conformation, which is more likely to protect them from instability at higher temperatures [75].

From a kinetic point of view, a very common feature of all low-temperature active enzymes is the reduction of activation enthalpy, accompanied by negative entropy, which reduces the exponential decline of the chemical reaction rate resulting from the reduction in temperature. It has been reported that the surface flexibility of low-temperature active enzymes is the source of cold adaptability [76]. For example, Isaksen discussed the role of protein surface mobility in determining the enthalpy centre equilibrium and subsequently proved that the surface of the enzyme is rigid through a single remote mutation of the key residue of protein water surface interaction. Therefore, cold-denatured trypsin can be transformed into a variant with medium-temperature characteristics without changing the amino acid sequence [77]. It has been suggested that the rigidity of the surface of low-temperature active enzymes may produce a higher activation enthalpy of the catalytic reaction with fewer negative entropy components. 

**Figure 2 ijms-24-08532-f002:**
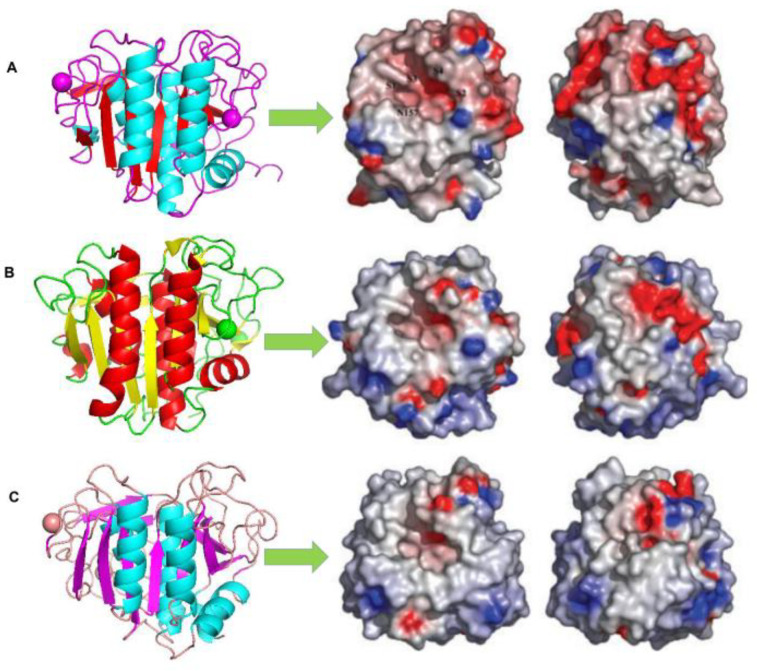
Comparison of the electrostatic surface potentials and crystal structures of (**A**) 1SH7, (**B**) 1IC6, and (**C**) 1THM [73]. On the right-hand side, the molecules have been rotated 180 about the y-axis. The approximate locations of substrate binding pockets, S1–S4 (nomenclature according to) and the oxyanion hole residue, N157, are labelled on the surface of the Vibrio proteinase (**A**) [78]. The positive potential is in blue and the negative potential is in red. The electrostatic surface potential was calculated using Delphi, and the graphical presentations were produced using Pymol [63].

## 6. Thermal Stability Modification of Cold-Adapted Proteases

The high hydrolytic activity of cold-adapted proteases against proteins at low temperatures is a crucial advantage for their commercial application. Nevertheless, the low thermal stability of cold-adapted proteases is a common disadvantage that hinders their industrial application. To address this limitation, we can improve the stability of the enzymes in a higher temperature range by site-directed mutation, directed evolution, and rational design of proteases. The basis for improving the thermostability of psychrotrophic proteases is primarily to increase disulphide and hydrogen bonds, salt bridges, and aromatic interactions to increase the affinity of the enzymes to calcium and modify autocatalytic sites. When discussing the relationship between the thermal stability modification of enzymes and flexibility, we also need to emphasise the location and distribution of flexible regions. It is necessary to consider the possibility of reduced enzymatic activity due to the reduced flexibility of some regions. On the other hand, in modifying the thermal stability of the flexible regions of enzymes, the current focus is mainly on the B-factor, which reflects the static flexibility of proteins. In wild-type enzymes, amino acid residues with higher B-factor values are more flexible and can be used as mutant sites. Molecular dynamics simulations at different temperatures revealed that localisation of flexible regions is better because the RMSF (a parameter that reflects atomic degrees of freedom) degrees of freedom of amino acid atoms increase with temperature in simulations at different temperatures, such as with transketolase (EC 2.2.1.1). Through simulation at different temperatures (300, 340, and 370 K), it was found that RMSF values of loop 6, loop 8, loop 15, loop 17, and loop 33 regions increased with increasing temperature and were higher than those at other positions in the structure. Finally, the t1/2 of the screened mutant A282P/H192P was increased three-fold (Figure 1) [79]. Pantoliano et al. found six amino acid mutations and subsequently combined them through random mutation and sequence homology, significantly enhancing BNP stability [80]. By deleting the calcium-binding ring of BNP and then conducting directional mutagenesis and selection, Strausberg et al. improved this enzyme’s stability 1000-fold under conditions of a high metal chelation [81]. Narinx et al. realised the importance of molecular structure stability by adding additional salt bridges and disulphide bonds, modifying the affinity between enzymes and calcium to subtilisin subt1 from *Bacillus antarctica* TA39, and modifying the calcium ligand T85D so that the thermal stability of the mutant reached the level of mesophilic subtilisin. Unexpectedly, the overall activity of the mutant was approximately 20-fold higher than that of medium-temperature subtilisin, thus indicating that the stability and specific activity of the enzyme were not systematically negatively correlated [82]. Siguroardottir et al. improved the thermostability of subtilisin protease from *Vibrio psychrophilis* through site-directed mutagenesis while also improving its catalytic activity [83]. Pan et al. showed that the disaccharide trehalose could effectively stabilise the structure of the protein, while the autolysis and activity loss of MCP-01 was significantly weakened after adding trehalose to the cold-adapted protease MCP-01 under the same conditions [84]. Furthermore, He et al. showed that trimethylamine oxide could improve the thermal stability of MCP-01, a new type of subtilise from deep sea cold-tolerant bacteria while retaining the hydrophilicity and conformational flexibility of cold-adapted enzymes [85]. Papain was immobilised on magnetic nanocrystalline cellulose (MNCC) using a new type of bio-based nanocomposite. In addition to its high stability in organic solvents, this protease has a higher optimal temperature and pH than free enzymes [48]. Activity and productivity at low temperatures can be improved by reducing the chemical modification of non-competitive substrate inhibition. Aspartic protease (EC 3.4.23) of the autocatalytic sites L205-F206 was replaced by non-conserved residues within 5A°, and it was found that the T_m_ value of the three-point process variant F193W/K204P/A371V increased by 10.7 °C [86].

It should be emphasised that in some previous reports, the low stability of psychrotrophic proteases at medium temperature was not a disadvantage for the local drug administration treatment of medical applications. However, psychrotrophic proteases have high activity and efficiency when applied locally at ambient temperature. Once the temperature reaches warmer conditions in the body, they lose activity and subsequently degrade, thus limiting the duration of enzyme activity to ensure that they only show controllable local therapeutic effects [13]. However, when it is required to extend the half-life or catalytic efficiency of cold-adapted proteases to obtain a longer exposure time, greater tolerance can be achieved by providing drugs in the inactive zymogen form (which is then activated in vivo) or reducing the antigenicity and immunogenicity by engineering the protease [69].

## 7. Production of Cold-Adapted Proteases

The industrial application of cold-adapted proteases requires the large-scale cultivation of microorganisms and the fermentation of enzymes. However, limited research has focused on optimising the production of cold-adapted proteases in microorganisms. Therefore, we believe that the production of cold-adapted proteases should focus on several aspects: (1) screening of high-yield microbial strains and (2) determining the optimal enzyme production conditions for microbial fermentation. Considering the poor stability and short autolysis of cold-adapted proteases, screening their production strains should consider the maximum activity and yield of the target protein in the relevant production process.

Several studies have shown that culture conditions promoting the production of proteases in the microbial production of enzymes are significantly different from those that promote the growth of microorganisms. For example, in a study by Shi et al., the optimal growth temperature for *Bacillus cereus* was found to be 25 °C, while the optimal production temperature of protease was 15 °C [58]. Most microbial proteases exist outside of cells, and the production environment temperature, pH, nitrogen and carbon sources in the culture medium, inorganic salts, and the concentrations of stirring and dissolved oxygen in the production process all have different degrees of influence on their production [69]. In a study by Shi et al., glucose was found to be the optimal carbon source for protease production by *B. cereus* cold nutrition, while the optimal pH environment for protease production was at an initial pH of 6.5–7.0 [58]. However, Wang et al. reported that the presence of carbohydrates such as glucose and sucrose inhibited protease production [43]. Kuddus et al. reported that the optimal nitrogen source for *Campylobacter luteum* to produce protease was skimmed milk and casein, with an optimal pH for enzyme production of 7.0. However, Dube et al. showed that the growth and protease production of different strains isolated from sediment samples of Antarctic lakes are different for casein, skimmed milk, bovine serum protein, and gelatin [51,87]. Vazquez et al. (2000) showed that when calcium chloride was present in the culture medium and its concentration (0–0.3 gL^−1^) increased, the protease production of *Stenotrophomonas maltophilia* also increased [88]. Each microorganism or strain has its own characteristics, physical chemistry, and nutritional requirements. Superior large-scale fermentation technologies, relative to traditional methods, which may still not yield the best results owing to high costs and being time-consuming, could be more conducive to the large-scale fermentation of psychrotrophic proteases. Wang et al. optimised the fermentation medium of cold-active protease in Colwellia sp. NJ341 using the response surface method and subsequently increased protease production approximately three-fold, reaching 183.21 U/mL [43]. Han et al. also optimised the mineral composition of the culture medium through statistical methods and, consequently, increased the activity of protease W-Pro21717 15-fold compared with that of non-optimised bottled fermentation through batch-fed fermentation [89]. BiałKowska et al. used Taguchi’s mathematical method and variance analysis to optimise the fermentation culture of Sporobolomyces roseus LOCK 1119 to maximise the biosynthesis of extracellular proteases [90].

Most cold-adapted microorganisms and their proteases are found in extremely cold environments. If the original psychrophilic microbial strains were used to produce cold-adapted proteases, large-scale culture and fermentation would not be possible due to the low temperature, anaerobic environment, extreme pH, and high salinity, among other strain culture environmental factors. The large-scale fermentation of psychrotrophic proteases can be improved by cloning the representative genes of psychrotrophic proteases in vitro using bioengineering technology and subsequently inserting the corresponding vectors into the normal temperature heterologous expression host. Currently, there are few studies on the heterologous expression of cold-adapted proteases, whose expression level is generally low. Macouzet et al. successfully heterologously expressed trypsin from *Tautogolabrus adspersus* in yeast, while Jónsdóttir et al. Pálsdóttir et al. actively expressed trypsin from the Atlantic Ocean in an E. coli expression system [91,92,93]. Kurata et al. cloned the cold-adapted protease gene acpI from deep sea *Alkalimonas collagenimarina* A40 and successfully heterologously expressed and purified it in E. coli. The expression amount was only 35.8 U/mg, which was significantly lower than that of the original strain [94]. Therefore, producing cold-adapted proteases highly expressed in the normal temperature expression host represents an important aspect of production research. Recent research has shown that the protease hydrolysis active site is artificially introduced into the esterase scaffold to facilitate the production of the protease esterase-integrated artificial enzyme EH1AB1. Compared with natural proteases, this artificial enzyme has a higher yield [95].

## 8. Medical Applications of Cold-Active Protease

As well-tolerated therapeutic drugs, proteases have been used in medicine for decades, and considering their high activity under low and moderate temperature conditions, cold-adapted proteases provide advantages such as being a unique therapeutic drug in different clinical applications. Presently, 24 types of protease drugs are approved by the US Food and Drug Administration (FDA), of which 5 are cold-adapted proteases, which usually have higher specific activity, low substrate affinity, and low-temperature adaptation than mesophilic proteases. For example, the catalytic efficiency of cod trypsin is 17-fold greater than that of bovine trypsin [8]. Proteases are generally thought to participate only in processes related to digestion, although their functions are more complex than digestion itself [96]. Proteases have been proven to be involved in regulating many cellular components, from growth factors to receptors, as well as processes including immunity, complement cascade, and blood coagulation. In addition to participating in homeostasis, the increase or imbalance in protease activity is also related to tumour growth and invasion [97].

Cold-active proteases are used in biopharmaceuticals since they participate in peptide synthesis at low temperatures. In vivo, peptides also participate in cell repair and metabolic process regulation and may be used as hormones, neurotransmitters, and growth factors. In vitro, they are often used as bioactive substances in drugs or their precursors. Peptide bonds can usually be formed in a stereoselective manner through proteases. Considering that cold-adapted proteases maintain activity at low temperatures (<20 °C), this is conducive to stable peptide synthesis [69]. A single protease can be used to synthesise small peptides (dipeptides and tripeptides), such as papain, chymotrypsin, and thermolysin, to synthesise carotid peptides, angiotensin, enkephalin, dynorphin, and the tripeptide Arg-Gly-Asp (a new drug for treating severe burns and skin ulcers). The synthesis of long peptides requires several proteases with different substrate specificities. For example, Kullmann used papain α-Cholecystokinin, which is synthesised by the chemical condensation of peptides obtained from chymotrypsin, aryl sulfatase, thermolysin, and aminopeptidase M [69].

Protease can effectively remove necrotic tissue and fibrin clots while also playing an important role in wound debridement and promoting tissue healing and regeneration. Mekkes et al. found that the Antarctic krill protease can effectively debride necrotic wounds and is better than normal saline in promoting wound recovery [98,99]. Cold-adapted proteases also have application potential in reducing periodontal disease and protecting gingival hair. Berg et al. conducted plaque removal experiments in vivo and in vitro using Antarctic krill protease and found that it could inhibit the adhesion of saliva-coated hydroxyapatite microorganisms and remove dental plaque, which has good application potential for plaque control [97]. Moreover, when applied to chewing gum formulations, protease can destroy the adhesion of bacteria to the dental membrane and significantly reduce gingival bleeding [100]. Regarding virus prevention, post-marketing research data of some products show that using cold-adapted enzyme oral spray, as an oral solution containing glycerine and cold-adapted cod trypsin, can reduce the incidence of colds [101].

Cold-adapted proteases have biotechnological applications in medical research. For example, cold-adapted proteases can be used to isolate single cells from organs and tissues at temperatures close to the freezing point. This method can better preserve gene expression patterns in vivo for single-cell analyses [102]. Similar evidence has also been reported by O’Flanagan et al. who found that single-cell RNA sequencing with cold-active proteases to isolate solid tumour tissue can minimise the conservative collagenase-related stress response [103]. However, the application of cold-adapted proteases in medicine has been limited by their heat sensitivity, alkalinity, and easy autolysis. The stability of cold-adapted proteases and the compatibility of active drugs can be further improved through the formation of protease nanoparticle complexes or chemical modification and enzymatic modification; furthermore, the potential application of proteases as drugs can be greatly expanded by improving their delivery mode through rational protease design [15]. Proteases can assist in extracting medical biopolymers and active substances or can be used in medical research. For instance, Domingo et al. [104] used proteases and hemicellulases to jointly extract alcohol-insoluble soluble dietary fibre, including inulin, pectin, and polyphenols, from artichoke industrial waste [105]. In addition, Karama et al. evaluated the antioxidant activity of flaxseed proteins hydrolysed by different proteases, while Dvoryakova et al. [106] found that protease from Castanea henryi still had high activity at 20 °C and can be used to develop various types of drugs for gluten intolerance.

## 9. Other Industrial Applications of Cold-Adapted Proteases

According to BCC research, the global enzyme market is expected to grow from 5.01 billion in 2016 to 6.32 billion in 2021 (Figure 3). In industrial enzyme statistics, protease usage accounts for approximately 60% of industrial hydrolases [55]. Among them, *B. subtilis* protease (EC 3.4.21.62, an alkaline serine endopeptidase) accounts for the largest market share and is used in the pharmaceutical industry, detergents, textiles, food and feed, leather, photography, cosmetics, and environmental remediation [107]. Through research and review of cold-adapted proteases, researchers have realised that cold-adapted proteases and cold-adapted microorganisms have broad prospects in terms of commerce and biotechnology, with some important characteristics of cold-adapted proteases making them widely used in biotechnology: (1) they are cost-effective since the specific activity of the enzyme is high; (2) they can catalyse reactions without requiring additional activation energy; (3) selective deactivation can be achieved through minimal heat input. In the future, more cold-adapted proteases will be used in industrial production instead of mesophilic proteases.

## 10. Cold-Adapted Proteases and Detergents

According to statistics, 25–30% of the enzyme market depends on the detergent industry, which is one of the successful applications of enzyme biotechnology [109]. The dirt on clothes mainly comprises protein, starch, and lipid components, among which protein dirt includes blood, egg stains, cocoa, grass, and human sweat, mainly from daily life or industrial environments. The service life of fabrics can be shortened by heating and beating to remove stains or dirt while adding mesophilic enzymes to detergents must be accompanied by heating, which may lead to fabric wear. Previously, people used to replace enzymes with strong detergent chemicals, but when released into the environment, these artificially added chemicals are difficult to degrade, resulting in water and soil pollution. With the progress of biotechnology and exploration of cold-adapted biological resources, to minimise energy requirements and maintain fabric quality, low-temperature active enzymes were first used in the detergent industry, not only improving economic value and efficiency but also reducing energy consumption and chemical use, thus reducing toxic by-product emissions and environmental pollution [110]. Commercially available low-temperature active enzymes include proteases and lipases, as well as amylases, cellulases, pectinases, and mannanases [39]. According to a market survey, the detergent industry will become one of the main consumers of hydrolases by 2024 [111]. Cold-adapted proteases make the overall structure of the molecule looser and easier to elute by hydrolysing the protein components in the stain. For example, proteases from *B. subtilis* WLCP1 can completely remove blood stains from fabrics after 20 min of treatment at 15 °C [1]. Through the compatibility test, the enzymatic characteristics of enzymes in detergent applications have been found to mainly include thermal stability, catalytic temperature range, pH stability, anionic surfactant stability, protein hydrolysis stability, chelating agent stability, chemical oxidant stability, bleaching stability, and a wide range of substrate specificity. Baghel et al. found that the protease produced by *B. subtilis* from cold environments showed abnormal stability in the presence of SDS, Tween 80, and Wheel had good pH and detergent compatibility at low temperatures [112]. However, detergents often work alongside surfactants and other chemical additives, and the stability of cold-adapted proteases may be affected. Improving the stability and prolonging the half-life of enzymes to ensure their high activity remains the main problem restricting the wide-scale application of cold-adapted proteases in detergents.

## 11. Cold-Adapted Proteases and the Food Industry

Proteins are abundant in food and provide amino acids for biological growth and health maintenance, while they are also used in combination with biotechnology to develop more applications. Temperature-adaptive proteases have also been used in the production process of traditional foods for a long time. However, using cold-adaptive proteases instead of temperature-adaptive proteases provides more possibilities for innovation and development in the food industry while avoiding bacterial growth and food corruption caused by mild environments when using temperature-adaptive proteases. Cold-adapted proteases from various microorganisms are widely used in the food industry, such as beer, bread and cheese industries, meat tenderisation, and functional food ingredients (active peptides, etc.) in the form of soluble protein hydrolysates [113]. Cold-adapted proteases can destroy the structure of intramuscular connective tissue and hydrolyse proteins, which is applicable to tenderising meat products. In addition, they can reduce TBA levels in meat products [40]. De Gobba et al. [114] extracted a cold-adapted protease from a *Saccharomyces cerevisiae* culture, which extensively degraded casein in cow’s milk (90%) after 24 h of hydrolysis at 5 °C and completely degraded it at 25 °C. The active peptide formed showed high angiotensin-converting enzyme inhibition and antioxidant activity. This discovery not only promoted the application of cold-active proteases in developing functional food ingredients but also provided a reference for the industrial application of biopharmaceuticals and cosmetics. However, in the field of food biotechnology, additional research is still needed to break through various bottlenecks, especially the high cost and low stability of low-temperature active enzyme separation and purification and the difficulty in developing cold-adapted microorganisms from cold habitats.

## 12. Other Applications

Considering their superior enzymatic properties and environmental advantages, cold-adapted proteases are also considered good feed additives for improving production performance in animal husbandry. Park et al. revealed that the cold-adapted protease of *Bacillus* sp. JSP isolated from Antarctica decomposes cutin, which can promote and support the processing of biomaterials composed of leather and poultry keratin waste [115]. Furthermore, Zambare et al. found that the protease from *P. aeruginosa* MCM B-327 had good application potential for the environmental protection processing of leather without the addition of buffers or chemicals [114]. Cold-adapted proteases also have prominent advantages in terms of environmental bioremediation and can be applied to the biodegradation and wastewater treatment of protein-rich wastes [62,88,112].

## 13. Expectations

Biosurfactants from cold-adapted organisms can interact with multiple physical phases, including water, ice, hydrophobic compounds, and gases, at low and freezing temperatures and can be applied to sustainable (green) and low-energy impact (cold) products and processes [14]. The relevant drivers for using cold-adapted proteases in industrial applications include the increased demand for consumer goods, the need to reduce costs, the consumption of natural resources, and environmental security. Cold-adapted proteases usually only need low temperature and pressure conditions to catalyse reactions. Because of their biodegradable and non-toxic properties, they are used as substitutes for hazardous chemical pollutants [109].

Currently, research on cold-adapted proteases mainly focuses on the discovery and identification, expression, and purification of new enzymatic genes, and the characterisation of related enzymatic properties. The huge technological gap between producing enzymes under laboratory conditions and obtaining the final commercialised product remains a problem in developing new biocatalysts. Some scientific challenges need to be addressed before the full potential of extreme enzymes can be realised. The stability of cold-adapted proteases is the biggest limiting factor in their application, but to date, there are relatively few studies on improving the stability of cold-adapted proteases from the perspective of molecular structure. Nevertheless, there have been many reports on the mechanism of cold acclimatisation enzyme cold tropism and thermal stability. The bottleneck of current transformation is finding a balance between enzyme activity, stability, and cold adaptability since most studies show that stability improvement is often accompanied by the loss of enzyme activity or changes in temperature adaptability. The elaboration of the cold adaptation mechanism and stability of psychrophilic proteases in this article may provide a theoretical basis for improving the thermostability of cold-adapted proteases. In addition, enzyme immobilisation technology is an attractive method to protect the stability and activity of enzymes. Accurate prediction and optimisation of the immobilisation strategy to adapt to implementing enzyme catalysis on a commercial scale are areas of concern.

Some studies successfully attempted to produce a small amount of recombinant cold-adapted proteases through recombination technology. However, the large-scale production of cold-adapted proteases is related to some complex factors. The most prominent disadvantage is their short half-life and autosolubility, making their production difficult under standardised industrial and temperature conditions. Therefore, for the expression of low-temperature enzymes, selecting the expression host and optimising the expression process are topics worthy of discussion. At present, most original psychrophilic microbial strains derived from cold-adapted proteases cannot be cultured on a large-scale due to the limitation of culture conditions. Normal temperature microbial hosts such as *E. coli*, *B. subtilis*, yeast, *Aspergillus niger*, etc. are relatively mature strains used in industrial microbial fermentation and can be used as expression hosts for cold-adapted protease production. However, there are few studies on the heterologous expression of cold-adapted protease locally and globally, and the reported cold-adapted protease activity is low. Therefore, screening and reforming more high-yield heterologous expression hosts of cold-adapted proteases and evaluating their safety and optimising their expression system are paramount. Screening cheaper nutrition sources, optimising the bacterial strain fermentation production process using response surface methodology, and finding the optimal production parameters to align with the economic surplus industrial production needs of cold-adapted proteases, which is also a valuable research area in the production process of cold-adapted proteases, are required.

## Figures and Tables

**Figure 1 ijms-24-08532-f001:**
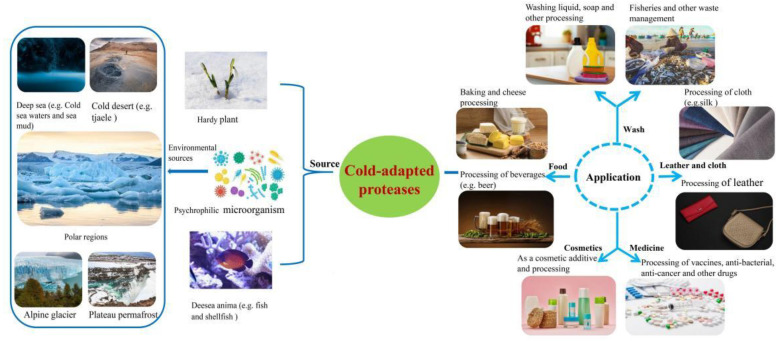
Source and application of cold-adapted protease (picture materials were purchased from Tuchacha and Biaoxiaozhi (https://chacha.so.com/copyright-home?srcg=home&src=query, accessed on 15 December 2022; https://www.logosc.cn/so/, accessed on 15 December 2022)).

**Figure 3 ijms-24-08532-f003:**
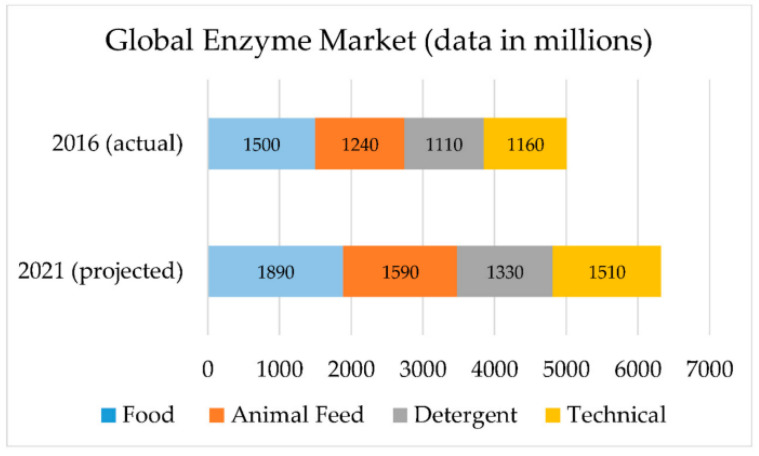
Global economy market in 2016 (**top**) and projected global economy market in 2021 (**bottom**). Figure from reference [108].

**Table 1 ijms-24-08532-t001:** Enzymatic properties of cold-adapted protease.

Source/Protease	Temp Optima (°C)	pH Optima	Molecular Weight (kDa)	Inhibitors	Activators	Refs.
*Alteromonas haloplanktis*	20	8.0–9.0	74–76	Leupeptin, PCMB	-	[57]
*Aspergillus ustus*	45	9.0	32	PMSF, DFP, Cu^2+^	-	[45]
*Bacillus subtilis* PAMC 26541	40	7.0–7.5	107	PMSF	K^+^, Na^+^	[50]
*Bacillus* sp. S1DI 10	10	8	40	EDTA, PMSF, Leucopeptin	Fe^2+^, Mn^2+^, Co^2+^, Twain 80	[41]
*Bacillus subtilis* WLCP1	15	10	38	PMSF	Ca^2+^, Cu^2+^	[49]
*Bacillus cereus*	20	9.0	-	EDTA, PMSF, Ca^2+^, Cu^2+^, K^+^	Co^2+^, Fe^2+^	[46]
*Bacillus cereus*	42	7.0–8.5	34.2	Mg^2+^, Mn^2+^	-	[58]
*Curtobacterium luteum*	20	7	115	EDTA, EGTA	Zn^2+^, Cr^2+^	[51]
*Colwellia* sp. NJ341	35	8.0–9.0	60	PMSF, Fe^2+^	-	[43]
*Colwellia psychrerythraea*	19	6–8.5	71	EDTA, Zn^2+^, Mn^2+^	Na^+^, Mg^2+^	[55]
*Candida humicola*	37	1–1.2	36	-	-	[60]
*Chryseobacterium* sp. IMDY	10	7.0–8.0	27	PMSF, Hg^2+^, Zn^2+^	Na^+^, Ca^2+^	[40]
*Escherichia freundii*	25	10.0	55	Iodoacetamide, SDS	-	[61]
*Flavobacterium* *psychrophilum*	24	6.0–7.0	62	Benzalkonium chloride, Na^+^, Phenanthroline,	-	[53]
*Pseudomonas* *lundensis*	40	10.4	46	EDTA, PMSF Cu^2+^, Zn^2+^, Hg^2+^, EDTA, SDS	Mg^2+^, Ca^2+^	[56]
*Halobillus* sp. SCSIO 20089	30	8	35	EDTA	Ca^2+^, Mg^2+^, Mn^2+^	[55]
*Leucosporidium* *antarcticum*	25	6.7–7.1	34.4	-	-	[54]
*Lysobacter* sp.	40	9.0	35	PMSF, EDTA, Zn^2+^	Ca^2+^, Mg^2+^, Ba^2+^, Na^+^, NH4^+^, isopropyl alcohol	[52]
*Penicillium chrysogenum* FS010	35	9	41	PMSF, DFP, Cu^2+^, Co^2+^	Mg^2+^, Ca^2+^	[42]
*Psychrobacter* sp. 94–6 PB	30	9	80	-	-	[39]
*Planococcus* sp. M7	35	10	43	PMSF, TNBS, EDAC, EDTA, Cu^2+^, Ni^2+^	Fe^3+^, Ca^2+^	[44]
*Pseudomonas* DY	40	10	25	PMSF, DFP, AEBSF	Ca^2+^, Mg^2+^	[43]
*Pseudoalteromonas arctica* PAMC 21717	30	9	37	LAS, SDS	Ca^2+^	[59]
*Pedobacter cryoconitis*	40	8.0	27	-	-	[62]
*Pseudoaltermonas*. sp.	30	8.0	47	PMSF, Chymostatin, Trypsin	-	[63]
*Pseudomonas* strain DY-A	40	10.0	-	EDTA, EGTA, SDS	-	[47]
*Stenotrophomonas* IIIM-ST045	15	10	55	Zn^2+^, Cu^2+^, Co^2+^	Mg^2+^, Mn^2+^, Ca^2+^	[64]
*Serratia marcescens*	40	6.5–8.0	58	Phenanthroline, PSFM, EGTA, DTT	-	[48]
*Sphingomonas* *paucimobilis*	20–30	6.5–7.0	-	DFP, PMSF, AEBSF	-	[65]
*Trichoderma* *Atroviride*	25	6.2	24	SDS, Urea	-	[12]

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
