# Peer review of "Cold-Adapted Proteases: An Efficient and Energy-Saving Biocatalyst"

_ijms, 2023, doi:10.3390/ijms24108532_

Round 1

Reviewer 1 Report

This review is devoted to cold-adapted proteases. The topic is not hackneyed and very interesting. However, I would advise to improve the text a little.

First, the authors should work on the English style. Sometimes there are sentences that are difficult to understand. For example:

Section 4: Organic reagents such as EDTA, PMSF, and DFP often inhibit the activity of cold-adapted protease [42–44], metal ions such as Hg2+, Cu2+, and Co2+ [40,46].

Section 5: nuclear magnetic resonance spectroscopy, site-directed mutagenesis, X-ray crystal diffraction, and freeze electron microscopy are effective meth-ods that can be used to understand the amino acid sequence, crystal structure, and function of enzymes. Additionally, the high structural homology of enzymes at different temperatures provides an opportunity to study the structural characteristics that may lead to their adaptation to different temperatures.

It is not clear what the authors wanted to say here.

Secondly, the text is replete with complex sentences, the two parts of which are separated by a strange combination of punctuation marks ".;". I guess this needs to be fixed.

Thirdly, Section 3. First paragraph about the classifications of proteases is somehow too descriptive, without specifics, or the example is not enough. Nothing is said about the specificity of proteases, cold-adapted proteases - what are they, for the most part?

Fourth, Section 4: Why is a protease with a temperature optimum of 60C considered cold-adapted, see Table 1, ref. [41]?

Fifth, Section 5: I think the authors should be clearer about the structural differences between cold-adapted and temperature-adapted proteases. It remains unclear to me how the surface with polar amino acids affects stability and adaptation to cold temperatures.

In general, the review needs to be structured more strictly.

Author Response

Itemized Replies to Review Comments

(Manuscript ID: ijms-2319959)

Reviewers’ comments

Reviewer: 1

Comments to the Authors:

This review is devoted to cold-adapted proteases. The topic is not hackneyed and very interesting. However, I would advise to improve the text a little.

  1. First, the authors should work on the English style. Sometimes there are sentences that are difficult to understand.For example:

Section 4: Organic reagents such as EDTA, PMSF, and DFP often inhibit the activity of cold-adapted protease [42–44], metal ions such as Hg2+, Cu2+, and Co2+ [40,46].

Section 5: nuclear magnetic resonance spectroscopy, site-directed mutagenesis, X-ray crystal diffraction, and freeze electron microscopy are effective meth-ods that can be used to understand the amino acid sequence, crystal structure, and function of enzymes. Additionally, the high structural homology of enzymes at different temperatures provides an opportunity to study the structural characteristics that may lead to their adaptation to different temperatures. It is not clear what the authors wanted to say here.

RESPONSE: Thanks for the valuable evaluations on our paper. We have tried our best to reorganize manuscript frame to make it easier for readers to get information. For the writing, format and other issues of the our manuscript, we have proofread and revised the full text of the writing, also have done seriouses English language polishing, relevant polishing proof will be uploaded. 

  1. Secondly, the text is replete with complex sentences, the two parts of which are separated by a strange combination of punctuation marks ".;". I guess this needs to be fixed.

RESPONSE: Thanks for the suggestion. We have corrected the punctuation ".;" in the manuscript correctly. In order to make the manuscript  are understood betterly.

  1. Thirdly, Section 3. First paragraph about the classifications of proteases is somehow too descriptive, without specifics, or the example is not enough. Nothing is said about the specificity of proteases, cold-adapted proteases - what are they, for the most part?

RESPONSE: Thanks very much for the valuable suggestion. As for the classification of cold-adapted protease in Section 3, we have provided a more detailed and comprehensive supplement to this part, including its specificity, examples of relevant classification, etc.

  1. Fourth, Section 4: Why is a protease with a temperature optimum of 60℃considered cold-adapted, see Table 1, ref. [41]?

RESPONSE: Thanks for the opinion. After careful verification, the protease in reference [41] does not belong to cold-adapted proteases in the strict sense, although it is classified as cold-adapted protease in Table II of reference [1]. Therefore, we have revised relevant parts in this manuscript.

  1. Fifth, Section 5: I think the authors should be clearer about the structural differences between cold-adapted and temperature-adapted proteases. It remains unclear to me how the surface with polar amino acids affects stability and adaptation to cold temperatures.

RESPONSE: Thank you very much for your valuable suggestions. The relationship between the polar or non-polar amino acids on the protein surface and the stability and temperature adaptability of the cold-adepted proteases, which are determined by the important interaction of the amino acid and water that makes up the protein surface. Thermophilic proteases enhance their stability by increasing the hydrophobic effect of the polar amino acids on the protein surface, while differences in the surface charge distribution or increases in the non-polar surface area are thought to be related to the low temperature adaptation of proteases in cold adapted proteases. Figure 2 in the manuscript clearly reveals the relationship between polar or non-polar amino acids on the surface of different protease and temperature adaptation. Of course, the relevant content is also supplemented in the manuscript.

Reviewer 2 Report

This is a very interesting field and this review can be a useful reference for further investigations. I have a number of comments on aspects that needed to be improved.

30 out of 117 references are articles published in 2018 or later. This means that about 75% of the references have been published more than five years ago. It should be useful to improved the updating of the review with more recently published articles.

The Authors use in different parts of the manuscript the concept of "temperature-adapted protease" as opposed to "cold-adapted protease". It appears confusing for the reader, as a cold-adapted protease is a protease that is adapted at the low temperature of a cold environment, so it is a specific case of "temperature-adapted protease", and not an opposed case.

"After long-term exposure to low temperatures, cold-adapted microorganisms have evolved corresponding cold-adapted mechanisms"
This concept is reported in different parts of the manuscript, and suggests that the adaption mechanism occurs after the long-term exposure to low temperatures, whereas it is more reasonable that adaption occurs during the long-term exposure to low-temperatures. I suggest to reformulate by removing the word "after" (or similar forms) from the sentences about this concept.

"non-covalent interactions (such as salt bridges, hydrophobic interactions, aromatic aromatic interactions, and main chain side chain and side chain hydrogen bonds) are weakened at low temperatures"
I disagree with this sentence, as there are differences among these interactions for the effects of temperature on their strength. Although a reference is indicated by the Authors at the end of the sentence, I recall an extensive literature on the subject.

I also disagree with the sentence that at low temperature the flexibility is increased, as it is the opposite extensively reported (lower temperature reduces conformational flexibility, higher temperature increases conformational flexibility). Probably, the concept is not clearly presented due to the lack of reference to the temperature at which the flexibility is considered to be increased or decreased. I think that a paragraph on thermal stability and flexibility must be included to better clarify the temperature range between the two melting temperatures of proteins, with the explanation of the classical graph showing deltaG of the unfolded - folded equilibrium vs temperature.

"Approximately 80% of the Earth's biosphere and 90% of the marine environment experience temperatures below 5 °C " appears in contrast with "Most parts of the Earth are at low and moderate temperatures"

"Recently, many psychrophilic enzymes, including psychrophilic proteases, have been excavated through different biomolecules and genetic engineering technologies, and these have good application potential for different industrial applications, including metagenomes, psychrophilic enzymes that have been obtained from various cold environments"
I suggest to revise the form, as it is not clear what it means "including metagenomes", probably it must be positioned otherwise in the sentence.

In Figure 1, text is too small and not readable, certainly it can be enlarged on screen but not in print copies.

Check and apply conventions for reporting the names of species in italics. It has been applied in the table, but it is absent in the main text.

Typos and punctuation must be revised in many cases:

- low-temperature protease dis not require additional heating - correct dis
- Protease is primarily used for treatment in four fields: Oral medicine - correct capital letter of Oral
- Arn ó rsd ó ttir et al. previously reported that - correct the Author's name

- In many cases there is a punctuation correction needed; here some examples of a period followed by a semicolon, check the entire manuscript ofr further cases:
to establish a low-carbon and sustainable green economy.;  
there was little interest in microorganisms that could grow at low temperatures.;
with a temperature below 0 °C throughout the year.;
an average temperature below 4 °C.;
fast method of gene mining.;
to cause conformational changes in the enzyme molecule.;
three-dimensional structure.;
how proteases become adapted to low temper-atures.;
and so on

Author Response

temized Replies to Review Comments

(Manuscript ID: ijms-2319959)

Reviewer: 2

Comments to the Authors:

This is a very interesting field and this review can be a useful reference for further investigations. I have a number of comments on aspects that needed to be improved.

  1. 30 out of 117 references are articles published in 2018 or later. This means that about 75% of the references have been published more than five years ago. It should be useful to improved the updating of the review with more recently published articles.

RESPONSE: Thanks for the valuable suggestion. On the basis of not changing the writing content of the manuscript, we have replaced and supplemented the references before 2018 as far as possible, such as [3], [4],[23],[24],[25],[26],[28],[29], and supplemented [117][118][119].

  1. The Authors use in different parts of the manuscript the concept of "temperature-adapted protease" as opposed to "cold-adapted protease". It appears confusing for the reader, as a cold-adapted protease is a protease that is adapted at the low temperature of a cold environment, so it is a specific case of "temperature-adapted protease", and not an opposed case.

RESPONSE: Thanks for the suggestion. The temperature-adapted protease in the manuscript is a clerical error, and it has been modified to mesophilic proteases. Mesophilic proteases are adapted to mesophilic environment, which were compared with cold - adaptive protease in the structure, temperature adaptability and application characteristics were compared with cold - adaptive protease.

  1. "After long-term exposure to low temperatures, cold-adapted microorganisms have evolved corresponding cold-adapted mechanisms"This concept is reported in different parts of the manuscript, and suggests that the adaption mechanism occurs after the long-term exposure to low temperatures, whereas it is more reasonable that adaption occurs during the long-term exposure to low-temperatures. I suggest to reformulate by removing the word "after" (or similar forms) from the sentences about this concept.

RESPONSE: Thanks for the suggestion. We have reviewed the expression of the sentence and have changed "after" to "during" in the sentence.

  1. "non-covalent interactions (such as salt bridges, hydrophobic interactions, aromatic aromatic interactions, and main chain side chain and side chain hydrogen bonds) are weakened at low temperatures"I disagree with this sentence, as there are differences among these interactions for the effects of temperature on their strength. Although a reference is indicated by the Authors at the end of the sentence, I recall an extensive literature on the subject.

RESPONSE: Thanks for the valuable evaluations on our paper. We agree with your point of view, and have consulted more relevant literature and conducted a stricter review on the content of the article. The effects of non-covalent bonds (such as salt Bridges, charge interactions, and main side chain and side chain hydrogen bonds) on protease adaptation to different temperatures were further elaborated: First of all, low temperature environmental conditions will not weaken the non-covalent bonding of enzymes, but after long-term cold-adaptation evolution, the structure of enzymes has better flexibility to adapt to low temperature environment and lower free energy required for catalytic reaction compared with mild thermophilic protease. However, the relationship between the number of non-covalent bonds and the temperature adaptation of enzymes is not determined at present. For example, the number of salt Bridges of cold-adaptive protease 1SH7, thermophilic protease 1IC6 and thermophilic protease 1THM was compared in reference [73]. The number of salt Bridges of 1SH7 and 1IC6 was the same, and the number of salt Bridges of the two was only two less than that of thermophilic protease 1THM. Although there have been many reported instances of salt Bridges and hydrogen bonds introduced in the modification of thermal stability enhancement of many enzymes, an important aspect of their contribution to protein stability lies in their location and distribution [73].The relevant expressions and arguments are supplemented in the manuscript.

  1. I also disagree with the sentence that at low temperature the flexibility is increased, as it is the opposite extensively reported (lower temperature reduces conformational flexibility, higher temperature increases conformational flexibility). Probably, the concept is not clearly presented due to the lack of reference to the temperature at which the flexibility is considered to be increased or decreased. I think that a paragraph on thermal stability and flexibility must be included to better clarify the temperature range between the two melting temperatures of proteins, with the explanation of the classical graph showing deltaG of the unfolded - folded equilibrium vs temperature.

RESPONSE: Thanks for the suggestion. We agree with your point of view, when discussing the relationship between thermal stability modification of enzymes and flexibility, we also need to emphasize the location and distribution of flexible regions. It is necessary to consider the possibility of reduced enzymatic activity due to the reduced flexibility of some regions. On the other hand, in the modification of the thermal stability of the flexible region of enzymes, the current focus is mainly on the B-factor which reflects the static flexibility of proteins. In wild type enzymes, amino acid residues with higher B-factor value are more flexible and can be used as mutant sites. MD simulations at different temperatures (molecular dynamics) localization of flexible regions is better because RMSF(a parameter that reflects atomic degrees of freedom) degrees of freedom of amino acid atoms increase with temperature in simulations at different temperatures, such as transketoolase (EC 2. 2. 1. 1) Through simulation at different temperatures (300K, 340K, 370K), it is found that RMSF values of loop6, loop8, loop15, loop17 and loop33 regions increase with the increase of temperature and are higher than other positions in the structure. Finally, the t1/2 of the screened mutant A282P /H192P was increased by 3 times (Figure 1) [120].

  1. "Approximately 80% of the Earth's biosphere and 90% of the marine environment experience temperatures below 5 °C " appears in contrast with "Most parts of the Earth are at low and moderate temperatures".

RESPONSE:Thanks for the suggestion.  "Most parts of the Earth are at low and moderate temperatures" is a misrepresentation and has been corrected in the manuscript.

  1. "Recently, many psychrophilic enzymes, including psychrophilic proteases, have been excavated through different biomolecules and genetic engineering technologies, and these have good application potential for different industrial applications, including metagenomes, psychrophilic enzymes that have been obtained from various cold environments"I suggest to revise the form, as it is not clear what it means "including metagenomes", probably it must be positioned otherwise in the sentence.

RESPONSE:Thanks for the suggestion. In the manuscript, we have modified "metagenomes" for a more accurate description to "microbial metagenomes in cryogenic environments".

  1. In Figure 1, text is too small and not readable, certainly it can be enlarged on screen but not in print copies.Check and apply conventions for reporting the names of species in italics. It has been applied in the table, but it is absent in the main text.Typos and punctuation must be revised in many cases:

low-temperature protease dis not require additional heating - correct dis

Protease is primarily used for treatment in four fields: Oral medicine - correct capital letter of Oral Arn ó rsd ó ttir et al. previously reported that - correct the Author's name

RESPONSE:Thanks for the suggestion very much. For some mistakes in the manuscript, we have revised them, please check.

  1. - In many cases there is a punctuation correction needed; here some examples of a period followed by a semicolon, check the entire manuscript ofr further cases:

to establish a low-carbon and sustainable green economy.;

there was little interest in microorganisms that could grow at low temperatures.;

with a temperature below 0 °C throughout the year.;

an average temperature below 4 °C.;

fast method of gene mining.;

to cause conformational changes in the enzyme molecule.;

three-dimensional structure.;

how proteases become adapted to low temper-atures.;

and so on

RESPONSE:Thanks for the suggestion. After reviewing the full text, we have corrected the error mark ".; ".

Reviewer 3 Report

The manuscript “Cold adapted protease: An efficient and energy-saving biocatalyst” is well structured and provides information on cold-adapted proteases. However, a few comments and suggestions of changes are included.

Title

The title of this work seems to be very general because it is mainly focused on microbial proteases. I suggest “proteases” instead of “protease” as well as in all manuscript.

Introduction

Page 1, 2nd line – I think “peptide” could be deleted.

Page 1, 3rd line – Please clarify which are “other small-molecule substances”.

Page 1, 6th and 7th lines – Please clarify “…century following this…”

Page 2, 2nd paragraph, 12th line – I think it is “did not require”. Please check.

Page 4, 2nd paragraph, 9th line –It is “shown” and not “shows”, I suppose.

Page 5, 1st paragraph, 6th line – Ceren et al. apparently is not mentioned in the References. Please check.

Enzymatic characteristics of cold-adapted protease

Page 5, 6th and 7th lines – Please clarify “…metal ions, surfactants and other compounds…”

Page 6, 7th line – It is kDa.

Page 9, antepenultimate line – It is “Kuddus”.

Page 10, 2nd line – It is “Vazquez”. Please check.

Page 10, 1st paragraph, 15th line – It is “BiaÅ‚kowskaPlease check.

Page 10, 2nd paragraph, 11th and 12th lines – Please check the quotations Jonsdottir and Palsdottir.

Page 14, 2nd paragraph, 13th line – I suggest replacing “enzyme” with “enzymatic”.

Page 14, 2nd paragraph, 14th and 15th lines – Please revise the sentence “…relationship… substances” for clarification.

Page 14, 2nd paragraph, 15th and 17th lines – Please also revise the sentence “The elaboration of the cold… protease” because it is not clear.

Page 14, 2nd paragraph, 18th line – The technology of immobilized enzymes is known at least from the 1970 decade.

Page 14, 2nd paragraph, 19th line – The word “inspection” doesn’t seem the most adequate. Please check.

Page 14 – The last sentence (“Screening and… protease”) of the 3rd paragraph is too long and not clear. Please check.

References

Page 16 – The reference 45, Chen et al. is not quoted in the manuscript. Please check.

Author Response

Itemized Replies to Review Comments

(Manuscript ID: ijms-2319959)

Reviewer: 3

Comments to the Authors:

The manuscript “Cold adapted protease: An efficient and energy-saving biocatalyst” is well structured and provides information on cold-adapted proteases. However, a few comments and suggestions of changes are included.

1.Title

The title of this work seems to be very general because it is mainly focused on microbial proteases. I suggest “proteases” instead of “protease” as well as in all manuscript.

RESPONSE:Thanks for the valuable evaluations on our paper. We agree with your suggestion to amend "protease" to "proteases" in the title.

  1. Introduction

Page 1, 2nd line – I think “peptide” could be deleted.

Page 1, 3rd line – Please clarify which are “other small-molecule substances”.

Page 1, 6th and 7th lines – Please clarify “…century following this…”

Page 2, 2nd paragraph, 12th line – I think it is “did not require”. Please check.

Page 4, 2nd paragraph, 9th line –It is “shown” and not “shows”, I suppose.

Page 5, 1st paragraph, 6th line – Ceren et al. apparently is not mentioned in the References. Please check.

Enzymatic characteristics of cold-adapted protease

Page 5, 6th and 7th lines – Please clarify “…metal ions, surfactants and other compounds…”

Page 6, 7th line – It is kDa.

Page 9, antepenultimate line – It is “Kuddus”.

Page 10, 2nd line – It is “Vazquez”. Please check.

Page 10, 1st paragraph, 15th line – It is “BiaÅ‚kowska”. Please check.

Page 10, 2nd paragraph, 11th and 12th lines – Please check the quotations Jonsdottir and Palsdottir.

Page 14, 2nd paragraph, 13th line – I suggest replacing “enzyme” with “enzymatic”.

Page 14, 2nd paragraph, 14th and 15th lines – Please revise the sentence “…relationship… substances” for clarification.

Page 14, 2nd paragraph, 15th and 17th lines – Please also revise the sentence “The elaboration of the cold… protease” because it is not clear.

Page 14, 2nd paragraph, 18th line – The technology of immobilized enzymes is known at least from the 1970 decade.

Page 14, 2nd paragraph, 19th line – The word “inspection” doesn’t seem the most adequate. Please check.

Page 14 – The last sentence (“Screening and… protease”) of the 3rd paragraph is too long and not clear. Please check.

RESPONSE: Thank you very much for your work. I am very sorry for the many mistakes in the manuscript. After careful examination of the whole manuscript, we corrected many errors in the manuscript.

  1. References

Page 16 – The reference 45, Chen et al. is not quoted in the manuscript. Please check.

RESPONSE: Thanks for the suggestion. We have cited the literature [45] in "3. Classification of cold-adapted proteases" of the manuscript, please review it.

Round 2

Reviewer 1 Report

The second version of the review manuscript on cold-adapted proteases is a really improved one. The authors made several additional explanations and corrected the previously indicated data. However, I must note the remaining sloppiness in the design of the text itself (duplicate punctuation marks, lack of spaces between words, unnecessary use of capital letters, etc.).  In addition, authors should ask someone with native English to re-edit the English style.

Author Response

Itemized Replies to Review Comments

(Manuscript ID: ijms-2319959)

Reviewers’ comments

Reviewer: 1

Comments to the Authors:

The second version of the review manuscript on cold-adapted proteases is a really improved one. The authors made several additional explanations and corrected the previously indicated data. However, I must note the remaining sloppiness in the design of the text itself (duplicate punctuation marks, lack of spaces between words, unnecessary use of capital letters, etc.).  In addition, authors should ask someone with native English to re-edit the English style.

RESPONSE: Thank you for your valuable evaluation of our paper and your work on my manuscript. For many errors in the details of our manuscript, we proofread and corrected the full text of the manuscript, and polished the English language carefully. Please refer to the following figure for the English to re-edit voucher.

Reviewer 2 Report

The manuscript has been improved and is suitable for publication

Author Response

Itemized Replies to Review Comments

(Manuscript ID: ijms-2319959)

Reviewer: 2

Comments to the Authors: 

The manuscript has been improved and is suitable for publication.

RESPONSE: Thank you very much for your review and evaluation of my manuscript.   

Best wishes.